# COMPRESSION VIA PRE-TRAINED TRANSFORMERS: A STUDY ON BYTE-LEVEL MULTIMODAL DATA

## ABSTRACT

Foundation models have recently been shown to be strong data compressors. However, when accounting for their excessive parameter count, their compression ratios are actually inferior to standard compression algorithms. Moreover, naively reducing the number of parameters may not necessarily help as it leads to worse predictions and thus weaker compression. In this paper, we conduct a large-scale empirical study to investigate whether there is a sweet spot where competitive compression ratios with pre-trained vanilla transformers are possible. To this end, we train families of models on 165GB of raw byte sequences of either text, image, or audio data (and all possible combinations of the three) and then compress 1GB of out-of-distribution (OOD) data from each modality. We find that relatively small models (i.e., millions of parameters) can outperform standard general-purpose compression algorithms (gzip, LZMA2) and even domain-specific compressors (PNG, JPEG 2000, FLAC) — even when factoring in parameter count. We achieve, e.g., the lowest compression ratio of 0.49 on OOD audio data (vs. 0.54 for FLAC). To study the impact of model- and dataset scale, we conduct extensive ablations and hyperparameter sweeps, and we investigate the effect of unimodal versus multimodal training. We find that even small models can be trained to perform well on multiple modalities, but, in contrast to previously reported results with large-scale foundation models, transfer to unseen modalities is generally weak.

## 1 INTRODUCTION

Strong predictive models can straightforwardly be turned into strong lossless compressors, e.g., via arithmetic coding (Pasco, 1977; Rissanen, 1976; Witten et al., 1987). Consequently, large pre-trained foundation models, such as LLMs, achieve very high data compression on their training distributions and beyond (Delétang et al., 2024). However, when factoring in these models' parameter count into the compression ratio, too large models actually perform worse. For this reason, large foundation models with parameter counts on the order of billions cannot compete with standard compression algorithms such as gzip (Deutsch, 1996) or LZMA2 (Pavlov, 2019). The goal of this paper is thus to investigate whether pre-trained vanilla transformers can achieve compression ratios that are competitive with standard algorithms across a range of data modalities. This places fairly tight constraints on the maximal model size, leading us to investigate families of relatively small transformers (with millions of parameters). Note that our aim is not to build a practical transformer-based data compressor, as the computational footprint (running time, memory, FLOPs) of even small models is far beyond standard compressors. Instead, studying compression via pre-trained models provides insight into the models' *learned* inductive biases, e.g., whether they are domain-general, how they depend on the training data composition, and whether there is transfer between modalities.

Recently, Delétang et al. (2024) stated that "language modeling is compression", pointing out that log-loss minimization is equivalent to optimizing a lossless compression objective. To illustrate this point, the authors used billion-parameter LLMs that were trained exclusively on text (Llama 2 from Touvron et al. (2023b) and Chinchilla from Hoffmann et al. (2022)) to compress 1GB of image and audio data from ImageNet (Russakovsky et al., 2015) and LibriSpeech (Panayotov et al., 2015), respectively. They found that these models compress better than gzip or LZMA2 and even domain-specific compressors such as PNG (Boutell, 1997) and FLAC (Coalson, 2008), but only when parameter counts are not being accounted for. To see if competitive performance is possible, they

Figure 1: Overview of our training and evaluation data pipelines. We consider three data modalities: text, images, and audio. From these modalities we create training data mixtures of 165GB that are either unimodal or multimodal. After pre-training transformers on each of these datasets, we evaluate their compression ratio (i.e., factoring in models' parameter counts) on each of the three modalities. If the corresponding modality has not been seen during training, we refer to the evaluation as 'out-of-modality', otherwise it is 'in-modality'. Importantly, our evaluation is always performed on out-of-distribution data (different from any of the training data sources), even when it is in-modality.

also trained small-scale transformers (up to 3.2M parameters) on 1GB of Wikipedia (Hutter, 2006), but found that these models were significantly worse at compressing images and audio data.

The obvious open question is whether small transformers pre-trained on large (multimodal) datasets can achieve competitive compression ratios across different modalities and whether there is transfer to unseen modalities, as observed in the large-scale model case. We therefore conduct an extensive empirical study where we train families of decoder-only transformers on 165GB of either text, image, or audio data and all combinations of the three. We then use these models (with frozen parameters, i.e., offline training) to compress 1GB of out-of-distribution (OOD) data from all three modalities (see Fig. 1). We also compare against transformers that are trained purely online, i.e., on the data stream that is being compressed (Bellard, 2019; 2021), meaning that storage or communication of the transformer weights for decompression is not required (unlike for our pre-trained models). These online transformers currently achieve state-of-the-art results on the Large Text Compression Benchmark (Mahoney, 2006). Overall we find that small pre-trained transformers achieve competitive compression ratios, as our best models consistently outperform domain-general and domain-specific standard compression algorithms and are on par with the online transformers from Bellard (2021).

**Main Contributions** We make the following key contributions:

- We conduct a large-scale empirical study (hyperparameter sweeps, ablations) on the compression performance of small transformers trained on raw byte sequences of text, image, and audio data (and all combinations), across various model- and dataset sizes.

- We are the first to show that small pre-trained transformers achieve better compression ratios than general-purpose and domain-specific compressors on 1GB of out-of-distribution data across different modalities, e.g., 0.49 on audio vs. 0.51 for Bellard (2021) & 0.54 for FLAC.

- We show that training on multiple modalities only slightly deteriorates the performance on each individual modality but significantly boosts the compression ratios on multimodal data, as long as all the evaluation modalities are part of the training data mixture.

- We demonstrate that small pre-trained transformers fail to beat standard compressors on unseen data modalities (i.e., modalities they were not trained on), meaning that there is only weak transfer to novel modalities (which is not the case for LLMs (Delétang et al., 2024)).

## 2 BACKGROUND

Compression and prediction are "two sides of the same coin" (MacKay, 2003). This fundamental duality stems directly from Shannon's celebrated lossless source coding theorem (Shannon, 1948), which states that there is a well-defined lower bound for encoding data from a probabilistic source. For any data sequence $x_{1:n} := x_1 x_2 \dots x_n \in \mathcal{X}^n$ of length $n$ from a finite alphabet $\mathcal{X}$ sampled from a source $\rho : \mathcal{X}^* \mapsto (0, 1]$, a lossless compressor $c : \mathcal{X}^* \mapsto \{0, 1\}^*$ assigns a code $c(x_{1:n})$, i.e., a sequence of bits, from which the original sequence is recoverable without loss of information. The goal is to minimize the expected length: $L_\rho := E_{x \sim \rho}[\ell_c(x)]$ by encoding rare sequences with more bits and frequent sequences with fewer bits. Shannon's source coding theorem states that the minimal expected length is lower-bounded by the Shannon entropy of the source: $L_\rho \geq H(\rho) := \mathbb{E}_{x \sim \rho}[-\log_2 \rho(x)]$.

If the source's statistics are unknown, good compression becomes a statistical modeling problem, i.e., good compression relies entirely on being able to predict well sequentially. For any predictor $\pi : \mathcal{X}^* \mapsto (0, 1]$ the expected coding length $L_\pi^\rho$ for data drawn from $\rho$ is at least the cross entropy:

$$ L_\pi^\rho \geq \mathbb{E}_{x \sim \rho}[-\log_2 \pi(x)] = \mathbb{E}_{x \sim \rho}\left[-\log_2 \frac{\pi(x)\rho(x)}{\rho(x)}\right] = H(\rho) + D_{\mathrm{KL}}(\rho || \pi) \geq H(\rho), $$

which is also lower-bounded by the Shannon entropy of $\rho$. A mismatch between $\pi$ and $\rho$ thus leads to an excess length given by their KL divergence, and minimal coding length (maximal compression) implies $\pi = \rho$ across the whole support of $\rho$. Accordingly, some AI researchers have argued that compressing well is fundamentally connected to intelligence (e.g., Chaitin's famous "Compression is Comprehension" (Chaitin, 2006); Rathmanner & Hutter (2011); Grau-Moya et al. (2024)), and that building universal compressors will accelerate AI development (cf. the Hutter prize (Hutter, 2006), an ongoing competition to compress (1GB of) human knowledge). The duality between compression and prediction has also led to the (algorithmic) information-theoretic formulation of universal prediction, i.e., Solomonoff induction (Solomonoff, 1964a;b; Li & Vitányi, 2019), one of two key ingredients for AIXI (Legg & Hutter, 2007; Hutter et al., 2024), the theory of artificial superintelligence.

Consequently, Delétang et al. (2024) argue that lossless compression performance lends itself as a domain-general metric for assessing any predictor's quality, including foundation models. They further emphasize that foundation models trained by minimizing log-loss (a.k.a., next-token prediction-error or cross entropy loss) are explicitly trained to minimize the expected coding length:

$$ \min_\pi L_\pi^\rho = \min_\pi \underbrace{\mathbb{E}_{x \sim \rho}[-\log_2 \pi(x)]}_{\text{"log loss"}} = \min_\pi \mathbb{E}_{x \sim \rho}\left[\sum_i -\log_2 \pi(x_i | x_{<i})\right]. \tag{1} $$

Note that the problem of constructing the actual codes that achieve (near) minimal expected code length given a predictor is largely solved in information theory, with gold-standard algorithms such as Huffman coding (Huffman, 1952), arithmetic coding (Pasco, 1977; Rissanen, 1976; Witten et al., 1987), or asymmetric numeral systems (Duda, 2009). The latter two compress strings online by iteratively converting them into a single binary number with increasing precision (see Delétang et al. (2024) for an illustration or Chapter 2 in Hutter et al. (2024)). Arithmetic coding is an example of an online compression algorithm since it only requires a single pass through the data and compresses on the fly (unlike offline compressors, such as Huffman coding, that require multiple passes through the data). Both our models and Bellard (2021), which we compare against, use arithmetic coding and compress online. However, the difference is that we pre-train our predictor, i.e., we perform *offline training* on a dataset and then freeze its parameters (non-adaptive arithmetic coding), whereas Bellard (2021) performs *online adaptation* of the model parameters on the data stream that is being compressed (adaptive arithmetic coding). As a result, and unlike our compressors, Bellard (2021) does not communicate the trained weights for decompression but only the model architecture and training algorithm (i.e., the model parameters do not need to be factored into the compression ratio).

## 3 RELATED WORK

**Compression Without Transformers** Using neural networks as predictors for lossless compression has been extensively studied, both in conjunction with arithmetic coding (Lehtokangas et al., 1993; Schmidhuber & Heil, 1994; 1996; Mahoney, 2000; Mikolov, 2012; Knoll, 2014; van den Oord &

Schrauwen, 2014; Cox, 2016; Schiopu et al., 2018; Goyal et al., 2019; Liu et al., 2019; Mentzer et al., 2019; 2020; Schiopu & Munteanu, 2020; Rhee et al., 2022) and with asymmetric numeral systems (Hoogeboom et al., 2019; Kingma et al., 2019; Townsend et al., 2019; Barzen et al., 2022). Neural networks have also successfully been employed in lossy compression, e.g., by overfitting tiny networks to individual data points and transmitting the model weights rather than the original data (Dupont et al., 2021; 2022; Chen et al., 2021; Ladune et al., 2023; Kim et al., 2023).

**Online Transformers**   Most of the above approaches use a separate training set to pre-train models that are then used to compress a test set. Alternatively, the model can also be trained from scratch on the data stream that is being compressed (Bellard, 2019; 2021; Goyal et al., 2020; Mao et al., 2022). The main advantage of these adaptive online compressors is that they are (quasi) parameterless (since they are initialized from scratch when compressing a new data stream), meaning that the model size does not explicitly affect the compression ratio, even for large models (though it implicitly affects the training performance, e.g., large models train more slowly meaning that larger chunks of the initial data stream are only weakly compressed). The transformer-based adaptive online compressor of Bellard (2021) is currently state-of-the-art on the Large Text Compression Benchmark (Mahoney, 2006), and our evaluation (in Section 5) shows that our best models are on par across all modalities.

**Pre-Trained Transformers**   Most closely related to our work is the line of research by Valmeekam et al. (2023); Delétang et al. (2024); Huang et al. (2024); Li et al. (2024); Mittu et al. (2024), which investigates lossless compression via arithmetic coding with pre-trained foundation models, i.e., the Llama models (Touvron et al., 2023a;b; Dubey et al., 2024) and Chinchilla (Hoffmann et al., 2022). Delétang et al. (2024), in particular, also report good compression rates on unseen modalities (LLMs trained only on text compress images and audio data well). However, these studies differ from our work as they do not take the model size into account for the compression ratios, except for Delétang et al. (2024), who report both "raw" and "adjusted" compression ratios and find that LLMs are not competitive in terms of adjusted (i.e., the actual) compression ratios. To the best of our knowledge, our paper is the first to systematically investigate the use of appropriately sized pre-trained transformers for multimodal lossless compression in a regime where competitive performance w.r.t. standard compression algorithms is possible. In this regime, our study is the most comprehensive in that it also investigates multimodal training and cross-modal transfer of pre-trained transformers.

## 4   METHODS

We now describe our experimental setup (with additional details, e.g., sweeps, in Appendix A).

**Baselines**   We compare to various standard compressors, both general-purpose, i.e., gzip (Deutsch, 1996) and LZMA2 (Pavlov, 2019), and domain-specific, i.e., FLAC (Coalson, 2008) for audio data and PNG (Boutell, 1997) and lossless JPEG 2000 (Skodras et al., 2001) for images. Both gzip and LZMA2 (which is used by the 7zip software) are based on Huffman coding (Huffman, 1952) and the Lempel-Ziv-Welch algorithm (Welch, 1984). We use the default parameters for gzip, LZMA2, and JPEG 2000, compression level 12 for FLAC, and instruct PNG to find the optimal encoder settings. We also compare to the online transformer from Bellard (2021), with the default v3.3 parameters, which is the current state-of-the-art on the Large Text Compression Benchmark (Mahoney, 2006).

**Models**   We focus on decoder-only transformers (Vaswani et al., 2017) with SwiGLU activations (Shazeer, 2020) and post-layer normalization. Unless otherwise noted, we use 8 heads, an embedding dimension of $64$, a context size of $4096$ (bytes), and sliding windows without overlap or memory (full details in Appendix A.3). We always train and evaluate the models with the same context size (i.e., $4096$ by default). We train our models with the Adam optimizer (Kingma & Ba, 2015) for 2.5 million steps with a batch size of 32, which, for 165GB of data, roughly corresponds to 2 epochs. Due to the duality of compression and prediction, we minimize the standard (sequential) log-loss (Eq. (1)) during training, which is a maximum-compression objective (see Section 2).

**(No) Tokenization**   Tokenization is a commonly-used, *domain-specific* pre-compression step to boost transformers' performance by increasing their vocabulary size in order to fit more information into their limited context window (Lester et al., 2024), i.e., increased information density at the cost

of increased entropy. However, since our goal is to be domain-general, we do not use tokenization and instead feed our models directly with byte streams (we still have to choose how to flatten images and how to sample audio signals, which are minimal domain-specific preprocessing steps).

**Evaluation**   To evaluate performance, we compute the compression ratio (lower is better):

$$\text{compression ratio} := \frac{\text{size of compressed data} + \text{size of compressor}}{\text{size of uncompressed data}}, \tag{2}$$

which accounts for the model size and is equivalent to the "adjusted compression rate" of Delétang et al. (2024). We always evaluate on 1GB of out-of-distribution data, i.e., size of uncompressed data = 1GB. As Delétang et al. (2024), we compute the size of the compressor by encoding the model weights with `float16` (2 bytes per parameter) since this level of quantization does not significantly affect performance (Tao et al., 2022) and is standard for model inference. As a result, our model sizes range from 0.8MB to 40.3MB. Note that, similar to Delétang et al. (2024), we do not compress the model parameters, since naive approaches (e.g., compressing them with gzip) do not significantly decrease the model size (only by around 7%, which corresponds to a decrease in compression ratio of only 0.002821 for our largest model). However, as a result, the compression ratio we report is an upper bound, which could be improved by (losslessly) compressing the parameters (though with limited room for improvement in our regime, even in the best case).

**Training Datasets**   A key point of our investigation is to evaluate how well pre-trained transformers can compress data from different modalities — both if the modality was or was not part of the training data (Fig. 1 visualizes our data collection process). We create three different unimodal training datasets with audio, images, and text data, and four multimodal training sets (Appendix A.1 describes the datasets in full detail). This yields seven pre-training datasets in total, each consisting of 165GB of data: (i) 165GB of audio; (ii) 165GB of images; (iii) 165GB of text; (iv) 82.5GB of audio and 82.5GB of images; (v) 82.5GB of audio and 82.5GB of text; (vi) 82.5GB of images and 82.5GB of text; and (vii) 55GB audio, 55GB of images, and 55GB text. By training our models on all seven training data mixtures, we can investigate *in-modality* and *out-of-modality* compression ratios. For example, for a model trained on the text dataset, the in-modality compression ratio can be determined by evaluating on text, while audio or image data provide out-of-modality compression ratios.

**Out-of-Distribution Evaluation Datasets**   To mimic the setting for which standard compression algorithms were developed (and thereby ensure a fair comparison), where the compressor is programmed with only minimal statistical assumptions about the data (with stronger assumptions for domain-specific compressors), we evaluate on unseen, out-of-distribution (OOD) datasets for each modality and not on in-distribution held-out datasets (as commonly done in machine learning). To do so, we create a single OOD dataset of 1GB for each modality (full details in Appendix A.2).

## 5   RESULTS

In this section, we present our extensive experimental evaluation (additional results in Appendix B). Unless otherwise noted, we report the best results over two hyperparameter sweeps (described in Appendix A.3): (i) over the model- and dataset sizes, and (ii) over the model- and context sizes.

**Small Transformers Can Be Domain-General Compressors**   Figure 2 shows the best compression ratio attained on each of the seven out-of-distribution evaluation datasets when training a model on each of the seven training data mixtures (we report the best-performing model from our two sweeps for each training-evaluation pair). We observe that transformers can achieve state-of-the-art in-modality compression ratios, regardless of the concrete composition of the training mixture, outperforming standard compression algorithms (even domain-specific ones) in all cases where all evaluation modalities are part of the training mixture. In these cases, transformers thus learn the prototypical statistical patterns related to that modality during pre-training. Importantly, by comparing models trained on unimodal vs. multimodal data, we observe that multimodal training only slightly decreases the compression performance compared to the unimodal models on their respective modalities (despite only having half or a third amount of data from that modality). This means that it is possible to trade off a small amount of performance on each individual modality to obtain a very strong domain-general compressor via multimodal training (the gray bar in Fig. 2).

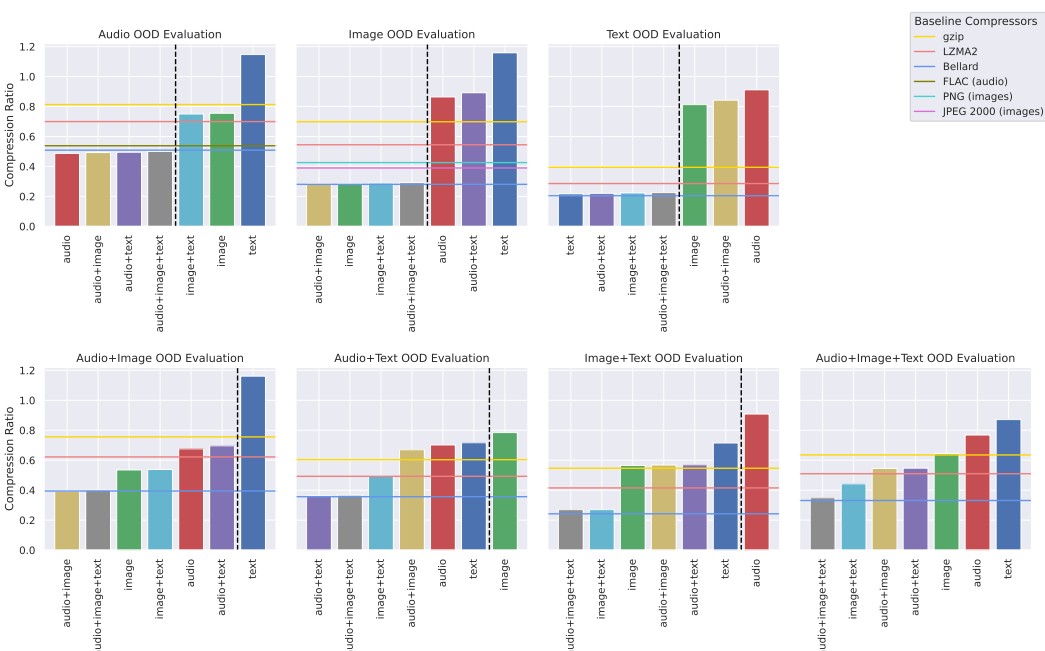

Figure 2: Small pre-trained transformers can be domain-general compressors (panels correspond to evaluation data mixtures, bars to training data mixtures). On every out-of-distribution evaluation data mixture, our method (i.e., the bars) outperforms standard compression algorithms (all horizontal lines except for 'Bellard') and is on par with Bellard's online adaptive transformers (the dark blue line) — as long as the evaluation modality was included in the training data mixture. For unseen modalities we observe very little cross-modal transfer (which is different from observations made with foundation models Delétang et al. (2024)). Unimodal training leads to models that are good for their respective modality, but multimodal training yields models that perform almost as well as the unimodal models across all their training modalities (despite seeing a lot less data per modality than the unimodal models), i.e., one can trade off a small amount of performance on each individual modality in return for a strong domain-general compressor via multimodal training (gray bar).

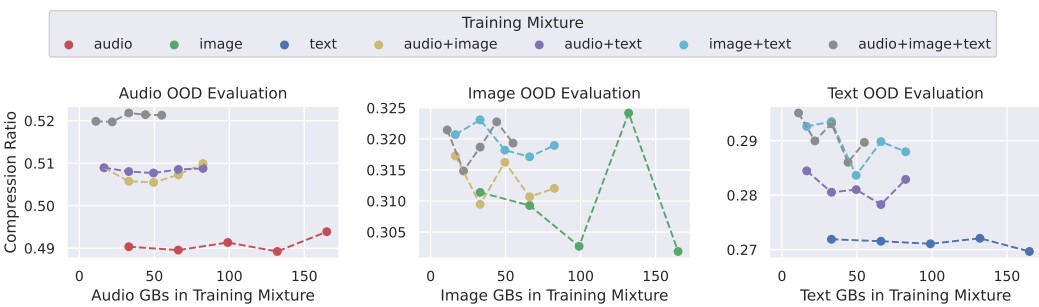

Figure 3: What you see is what you get. Each panel visualizes the compression ratios for one of our modalities when training models on varying dataset mixtures and sizes. Although one can replace a large proportion of a unimodal training dataset with a multimodal training mixture and not incur a significant loss on the original modality, transformers (at our tested model sizes) do not exhibit improved transfer from the out-of-modality data (i.e., the multimodal models are worse than the unimodal ones, even when trained on much more data from that particular modality). The upshot is that the multimodal training data does not hurt much (note the scale of the y-axis), but leads to significantly improved multimodal compression performance as shown in Fig. 2.

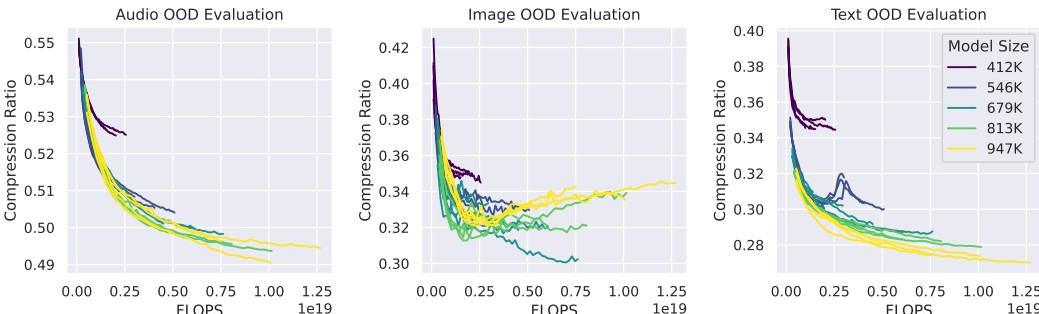

Figure 4: Simultaneously scaling training dataset and model size (for unimodal training- and evaluation data). The colors indicate the model size, and the lines correspond to different dataset sizes (20%, 40%, 60%, 80%, and 100%). We always train for 2 epochs, regardless of dataset size, i.e., smaller datasets require fewer FLOPS. As expected, increasing the number of parameters and the dataset size boosts compression (at the cost of increased training FLOPS). Note that our out-of-distribution evaluation makes models more prone to overfitting, as seen, e.g., for our largest models on images, making scaling more complex than traditionally observed LLM scaling laws.

**What You See Is What You Get** While Fig. 2 shows that substituting half or two thirds of the training set with data from other modalities only leads to a small performance loss compared to the unimodally trained models, it is unclear whether simply training on a smaller amount of unimodal data (i.e., decreasing the unimodal training dataset size to, e.g., 82.5GB and not substituting 82.5GB with data from another modality) would give the same performance, or whether there is some transfer between modalities (as suggested by Mirchandani et al. (2023)) that compensates for the smaller amount of data per individual modality. To investigate this, we run an ablation where we subdivide each of our seven training sets into 5 different sizes: 20%, 40%, 60%, 80%, and 100% of the respective dataset (uniformly subsampled). We train a series of models (sweeping over their number of layers; see Appendix A.3) on each dataset mixture and each dataset size, and then evaluate as before. Figure 3 shows that, for our models and datasets, there is little transfer between modalities. For all cases of audio, text, and (less clearly) images, it is better to train on a smaller unimodal dataset to get the best unimodal performance, as opposed to training on a much larger multimodal dataset. For example, training on a pure text dataset of 33GB (20% of 165GB) outperforms training on a dataset consisting of 82.5GB (i.e., more than twice as much) text and of 82.5GB images/audio.

**Scaling Analysis** Since there is a non-trivial relationship between model size and dataset size, we perform a scaling analysis on both of these factors (details in Appendix A.3). Figure 4 shows trends akin to the scaling laws observed for LLMs (Kaplan et al., 2020), which state that better prediction (in our case compression) is only possible by scaling both models and datasets, in a particular way. Note that, different to traditional scaling laws for models trained on internet-scale datasets, the distribution shift in our evaluation makes it easier for the model to overfit to the training distribution. However, as the number of parameters and the training flops of our small models increase, the adjusted compression ratio improves, eventually beating standard compression algorithms. We do observe gradual overfitting on the image dataset for our models trained only on images. However, this phenomenon can be mitigated by including other modalities in the training mixture (see Fig. A1).

**Model Size vs. Context Size** The previous two experiments investigated the impact of training dataset size and model size, which revealed a complex, "scaling law"-like, relationship between the two factors and the overall training budget in FLOPS. In this experiment, we investigate the impact of the length of the context window. Since the context window length has a large impact on the overall FLOPS footprint (attention scales quadratically with the input sequence length), we also vary the size of our models to explore whether there is a sweet spot in terms of training compute budget allocation (details in Appendix A.3). Fig. 5 shows that the optimal trade-off strongly depends on the data modality. The top performing models for text have a context window less than or equal to 2048 bytes, indicating that short term dependencies are more important than long ones in this case. For images, the best compromise overall is to choose a larger context window of 8192, which means

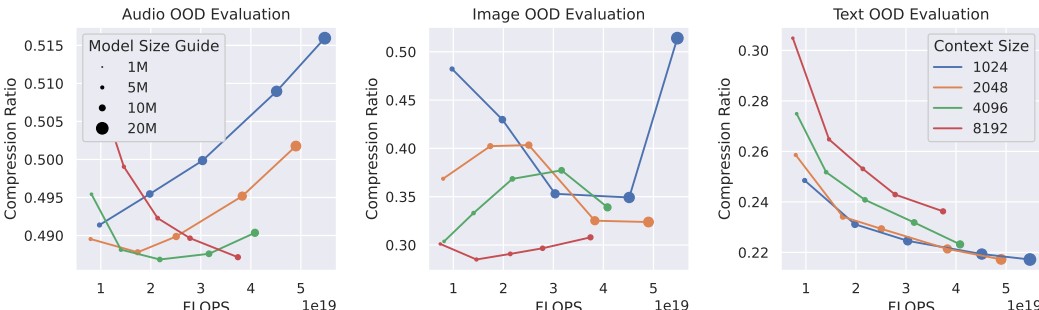

Figure 5: Relationship between context- and model size. Given a certain training compute budget (in FLOPS), one can either increase the context size (measured in bytes) or the model size, leading to a non-trivial trade-off. Our results show that this trade-off is highly modality-dependent (also note the different scales on the y-axis, meaning that the magnitude of the effect varies significantly with modality). For text, shorter context sizes and larger models are beneficial (indicating the importance of short-term dependencies for our data and model scale). For images, larger context is generally beneficial, which makes sense, given that a single image consists of $512 \cdot 512 \cdot 3 = 786432$ bytes, which far exceeds our models' contexts, i.e., models with larger context can process larger fractions of an image. Finally, for audio data the relationship is complex with intermediate context length and larger models performing better (though the reverse is true for short context length).

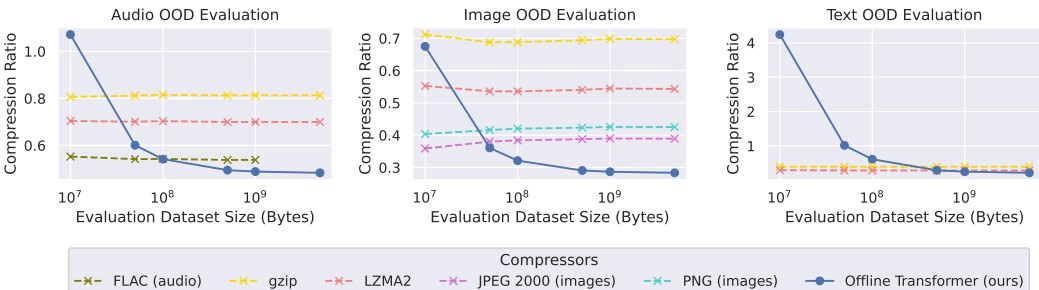

Figure 6: Compression ratio vs. evaluation dataset size. According to Eq. (2), the numerator of the compression ratio consists of the size of the compressed data *and* the size of the compressor. For standard compressors (e.g., gzip), the size of the compressor (a few thousand lines of code) is negligible given sufficient evaluation data (i.e., the compression ratio is unaffected by the evaluation dataset size). However, for neural compressors trained offline (i.e., where the size of the compressor is dominated by the model parameters), the compression ratio improves with increasing data since the model size has decreasing influence. Moreover, if the model is equal to or larger than the evaluation dataset size (e.g., 500M parameters and 1GB of data), one cannot achieve a compression ratio $< 1$.

decreasing the model size. For audio data, the trade-off is even more complex. Overall these results highlight the difficulty of tuning architectures to achieve best performance across many modalities.

**Evaluation Dataset Size**  Figure 6 visualizes the relationship between the compression ratio and the evaluation dataset size for all three modalities and our best-performing model (as determined on the standard 1GB of OOD data in Table A1). For offline (i.e., pre-) trained neural compressors, the model parameters have to be factored into the compression ratio, which means that their compression performance will improve with increasing evaluation data (as long as the model generalizes well to the additional data). In contrast, the size of standard compressors is negligible compared to the amount of evaluation data, which means that their compression ratios are largely unaffected by the evaluation dataset size. Note that FLAC cannot losslessly compress more than $\sim 4.2$GB of data.

**Sliding Window** In all experiments so far we used a sliding window without overlap to process the evaluation byte streams, i.e., we completely fill a whole context window, process it, and then slide it forward by the size of the context window to process the next chunk of data. This means that bytes early in the context window are not conditioned on a lot of data (in theory, conditioning on more data should help with prediction and thus compression, which may well be exploitable by transformers' in-context learning abilities (Brown et al., 2020; Genewein et al., 2023; Ge et al., 2024)). However, sliding the context window with more overlap requires more forward passes to process the same amount of data, which significantly increases the computational cost with increasing overlap. With no overlap processing 4096 bytes with a context window of 4096 takes a single forward pass. In the most extreme case of maximal overlap it would take 4095 forward passes, where the context window is moved by a single byte each time (though each prediction could be conditioned on the full 4095 preceding bytes). In our final experiment, we investigate the effect of different overlaps between context windows. Figure 7 shows that for our data and model sizes, increasing the overlap window (for a context length of 4096) has relatively little effect. The strongest effect is observed for image data, which makes

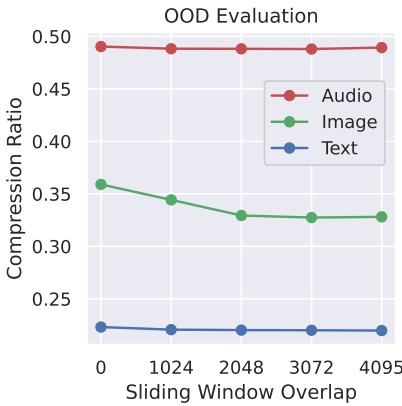

Figure 7: Impact of the sliding window overlap (for unimodal training and evaluation). Overlapping context windows only marginally improve the performance (most significantly for images) in our experiments but come at a huge cost in terms of computational efficiency.

sense given that 4096 bytes only corresponds to a small fraction of an image and there are obvious long-range dependencies between channels of the same image. Beyond an overlap of 2048 we do not see much benefit of further increasing the overlap window in our experiments.

# 6 DISCUSSION

The main goal of our work is to investigate whether pre-trained transformers can be competitive with standard compressors, even when taking their parameter size into account. In contrast to previous work, this places our models into a relatively small regime, where it is unclear whether models will learn well from large datasets at all and have non-trivial out-of-distribution and cross-modality transfer. This could partly be countered by training larger models and then subsequently compressing the model parameters themselves. We chose not to do this in our case since naive lossless compression of model parameters leads to a $10\%$ reduction at best (see Table A3), and even best-case scenarios would only lead to marginal improvements in compression ratio given the size of our largest models. For very large (e.g., foundation) models, compressing weights to achieve competitive compression ratios may be interesting, though it will be necessary to use lossy weight compression techniques (Tao et al., 2022), which lead to non-trivial trade-offs between high (lossy) compression and maintaining strong predictor performance, i.e., the two summands in the numerator of Eq. (2). Exploring these trade-offs is an interesting direction for future research but beyond the scope of our work. Another way to allow for larger models would be to simply evaluate on a larger test set. We deliberately chose to use 1GB of test data as a regime where standard compression algorithms are hard to beat. Additionally, evaluations on larger test data, and in settings where model parameters are not taken into account have previously conducted (Delétang et al., 2024; Valmeekam et al., 2023; Li et al., 2024) (where significant amounts of cross-domain transfer have also been found, unlike in our experiments).

Note that, similar to Xue et al. (2022), we do not use a tokenizer, which has two reasons. First, tokenizers are typically pre-trained per modality, and we want to rule out bad cross-modality transfer resulting from a bad tokenizer. Second, tokenization acts as a pre-trained "pre-compression" step (Delétang et al. (2024) make a similar comment). This pre-compression increases information density in the context window at the cost of increasing entropy, which can make the prediction problem harder: Lester et al. (2024) even show that when using a strong neural-based pre-compressor (together with arithmetic coding) to train LLMs, training performance can collapse catastrophically.

**Limitations** All our claims regarding the universality of our compressors (or the lack thereof) are limited to the model size regime and the particular modalities and datasets we studied. We cannot rule out that there are cases where even in-modality transfer is weak (e.g., when using another out-of-disribution image evaluation dataset with very different statistics), or that there may be cases of non-trivial cross-modal transfer (which we have not observed). We did not investigate transfer learning approaches to improve the out-of-modality performance of our neural compressors, but we consider this an interesting avenue for future work. Similarly, our claims regarding outperforming standard compression algorithms are limited to our experiments. We cannot rule out that there are datasets (such as spreadsheet data, or code, which, technically, are both text) where no pre-trained transformer outperforms, e.g., LZMA2 (in fact, we think its plausible that such datasets can be constructed synthetically). Moreover, we can also not rule out that other, more sophisticated architectures (e.g., Perceivers (Jaegle et al., 2021)), would outperform our models, and we consider investigating the optimal neural model architecture for lossless compression an interesting direction for future research. Finally, note that the goal of our study is not to build a practical transformer-based universal compressor to compete with standard compressors in terms of computational footprint. As Table A2 shows, our models are orders of magnitude slower for encoding data (and have significantly larger memory- and FLOPS-demands), and they are about three times slower than Bellard's online adaptive transformer. This is only the forward-pass cost, which can be done for a whole context window at once (without overlap). If our models were used do decode, which has to be performed token-by-token to obtain the correct conditioning, our running time demands would be even worse, making our models clearly uncompetitive in that sense.

# 7 CONCLUSION

In this paper we have shown that it is possible to use pre-trained vanilla transformers as competitive "zero-shot" compressors on out-of-distribution evaluation data, where competitive means achieving better compression ratios than both domain-general and domain-specific standard compression algorithms. We found this to be true for text, images, and audio data, and for all possible combinations of the three — but only as long as the corresponding modalities have been seen during training. We further found that, despite their relatively small size, our models have the capacity to train on multiple modalities, and then compress these well, without losing much performance compared to a purely unimodal model. On the other hand, we found that even multimodal training does not lead to the emergence of a universal compression ability that would yield strong compression performance on unseen modalities. This is in contrast to observations made by Delétang et al. (2024) on LLMs and indicates that there is a qualitative difference between small and (very) large models, even when the small models are trained on large amounts of data. Overall our results suggest that small transformers can be pre-trained to recognize and exploit statistical regularities on par and even better than hand-crafted standard compressors and current state-of-the-art adaptive online neural compressors, but we do not observe the emergence of a general compression ability with our model sizes.

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

## A   EXPERIMENTAL DETAILS

### A.1   TRAINING DATA SOURCES

We source all of our data from the following open-source TensorFlow datasets (Pot et al., 2019):

**Text**   Since most of TensorFlow's text datasets are quite small, we concatenate the following five datasets into a single collection of 165GB: (i) *Wikipedia* (Wikimedia, 2023), the filtered UTF-8 encoded text from an XML dump from 2023-06-01, containing all languages but predominantly English and western languages (113.9GB); (ii) *PG-19* (Rae et al., 2020), books from the Project Gutenberg, also encoded in UTF-8 (9.4GB); (iii) *Big Patent* (Sharma et al., 2019), a dataset of patents in English (30.2GB); (iv) *Scientific Papers* (Cohan et al., 2018), from arXiv and PubMed, containing the raw text including the LaTeX code (8.1GB); and (v) *Natural Instructions* (Mishra et al., 2022; Wang et al., 2022), tasks formulated in English covering different domains and lanugages (4.1GB).

**Image**   We collect a subset of 165GB of the ImageNet dataset (Russakovsky et al., 2015), uniformly sampled across the 1000 classes, which contains 14 197 122 annotated images (of varying resolutions) from the WordNet hierarchy. We decode the images into RGB arrays (three `uint8` channels), flatten them, and concatenate them into a byte stream of flattened images. As a consequence, we ignore image boundaries when sampling from this data source (i.e., sequences are not guaranteed to start or end at the start or end of an image).

**Audio**   We create a subset of 165GB from the Common Voice dataset (Ardila et al., 2020), a multilingual dataset of voice recordings. We downsample the dataset from 48 kHz to 16 kHz and encode the waveform as `int16`, i.e., with two bytes per sample. As for images, we concatenate all individual audio samples into a single byte stream. Accordingly, there is no guarantee that a sequence sampled from our dataset starts or ends at the beginning of a recording.

### A.2   OUT-OF-DISTRIBUTION EVALUATION DATA SOURCES

We source all of our data from the following open-source TensorFlow datasets (Pot et al., 2019):

**Text**   We consider a 1GB subset of the *Reddit* dataset (Völske et al., 2017), which contains 3.8 million Reddit posts encoded in UTF-8.

**Images**   We create a 1GB subset of the CelebA HQ dataset (Liu et al., 2015) with a resolution of $512 \times 512$. We process the images in the same way as for our image training set, i.e., flattening and concatenation, and we subsample uniformly across classes of CelebA.

**Audio**   We use 1GB from the *LibriSpeech* (Panayotov et al., 2015) dataset, which contains roughly 1000 hours of English speech data derived from audiobooks that have been segmented and aligned in the LibriVox project. The data is already in 16kHz (with a sample size of 2 bytes), and we simply concatenate samples into a single byte stream.

**Multimodal Evaluations**   For our evaluations on multimodal data, we use the unimodal evaluations on 1GB of data as described above and average the results accordingly (both for our models but also all standard compression algorithms, and Bellard's online adaptive transformer), either over two or three evaluations depending on the evaluation mixture composition.

### A.3   SWEEPS

**Model Size vs. Dataset Size**   The experiment to investigate the impact of training dataset- and model size, with results shown in Fig. 4, used the following model parameters. Dataset sizes were 20%, 40%, 60%, 80%, and 100% of the full 165GB for each training set mixture (uni- and multimodal). All models used a context size of 4096, 8 attention heads per layer, a widening factor of 4 and the number of layers was either 2, 4, 6, 8, or 10. Models were trained with a batch size of 32. The learning rate was $1 \times 10^{-4}$, and a sinusoid positional encoding was used.

Table A1: Best compression ratios for each compressor. This table shows the same results as Fig. 2 but as precise numerical values to facilitate detailed comparison.

| Evaluation Modality | Out-of-Distribution Compression Ratio | | | | | | |
|---|---|---|---|---|---|---|---|
| | **Ours** | **Bellard** | **gzip** | **LZMA2** | **FLAC** | **PNG** | **JPEG 2000** |
| Audio | **0.487** | 0.509 | 0.813 | 0.699 | 0.538 | - | - |
| Image | 0.285 | **0.281** | 0.698 | 0.545 | - | 0.426 | 0.390 |
| Text | 0.217 | **0.204** | 0.394 | 0.286 | - | - | - |
| Audio + Image | **0.393** | 0.395 | 0.756 | 0.622 | - | - | - |
| Audio + Text | 0.362 | **0.357** | 0.604 | 0.493 | - | - | - |
| Image + Text | 0.270 | **0.243** | 0.546 | 0.415 | - | - | - |
| Audio + Image + Text | 0.349 | **0.331** | 0.635 | 0.510 | - | - | - |

**Model Size vs. Context Size** Fig. 5 in the main paper shows the relationship between context length and model size. For this experiment we performed a large-scale sweep with the goal of covering a good range of training FLOPS budget with models that make various trade-offs between model size and context length (given the same model size, compute demand increases with increasing context length). The main question was whether there is a qualitatively similar relationship across parameters, and whether there is a clear sweet spot — see the main paper for results and discussion. For our sweep we used the same model parameters as in the previous paragraph (the training data size was always at $100\%$) and sweep over the following four context sizes (with training batch size in brackets): $[1024\ (128), 2048\ (64), 4096\ (32), 8192\ (16)]$. For each context size we train five models (XS, S, M, L, and XL) on all three unimodal datasets, respectively. Each model has a different combination of embedding dimension and number of layers for each different context size. The XS models have embedding dimensions $[112, 96, 80, 64]$ and numbers of layers $[11, 7, 5, 3]$ for the different context sizes respectively (i.e., wider and deeper models for shorter contexts and more narrow and more shallow models for long context size). The S models have embedding dimensions $[192, 160, 112, 96]$ and numbers of layers $[10, 8, 6, 4]$. The M models have embedding dimensions $[224, 192, 144, 112]$ and numbers of layers $[12, 9, 7, 5]$. The L models have embedding dimensions $[272, 240, 176, 144]$ and numbers of layers $[13, 10, 8, 5]$. The XL models have embedding dimensions $[320, 304, 240, 160]$ and numbers of layers $[12, 9, 7, 6]$. The main goal with these settings is to create families of models that have roughly the same demand in terms of FLOPS (iso-FLOPS) but very different trade-offs in terms of model- and context size.

### A.4    COMPUTATIONAL RESOURCES

We trained every model on 16 NVIDIA A100 GPUs from our internal cluster. We trained 315 models in total, yielding a computational footprint of 5040 A100s. We ran Bellard's code on an NVIDIA GeForce RTX 4090 GPU with a 24-core Intel i9-13900KF CPU @ 3Ghz.

## B    ADDITIONAL RESULTS

### B.1    COMPRESSION RATIOS

Table A1 shows the optimal compression ratios that each of the compressors achieve on all of the different evaluation modalities (note that all evaluations are on out-of-distribution data). The same values as shown in Fig. 2 in the main paper and given here as precise numerical values for completeness.

### B.2    RUNNING TIMES

Table A2 shows the wall-clock running times in seconds for compressing 1GB of data from each of the three modalities for our models, Bellard's online adaptive transformer (Bellard, 2021), and the standard compression algorithms used in our work. As the table clearly shows, our models and Bellard's model are orders of magnitudes slower (let alone the increased computational demand and

Table A2: Running times to compress 1GB of data for all compressors used in our study. Note that we use the best model per modality, which have different sizes and thus different running times.

| Evaluation Modality | Running Times [s] | | | | | | |
|---|---|---|---|---|---|---|---|
| | Ours | Bellard | gzip | LZMA2 | FLAC | PNG | JPEG 2000 |
| Audio | 305 609 | 101 178 | 55 | 524 | 169 | - | - |
| Image | 222 065 | 103 391 | 47 | 436 | 174 | 495 | 99 |
| Text | 452 355 | 100 657 | 102 | 881 | 184 | - | - |

Table A3: Compression ratios for model parameters. We losslessly compress the trained model parameters with standard compressors. For each modality we choose the best-performing model. As is shown, the maximal compression is $11\%$, which would affect the overall compression ratio on the corresponding evaluation data only very marginally.

| Evaluation Modality | Model Parameter Compression Ratio | |
|---|---|---|
| | gzip | LZMA2 |
| Audio | 0.93 | 0.90 |
| Image | 0.93 | 0.90 |
| Text | 0.92 | 0.89 |

GPU requirements). Note that running times for our models differ, because we pick the best model per modality, which are models of different sizes.

### B.3 COMPRESSING MODEL PARAMETERS

Throughout our paper we report compression rates that take uncompressed model parameters into account. As discussed in the main paper, compression ratios could be improved by also compressing model parameters. However, as Table A3 shows, naively compressing model parameters with a lossless compressor does not lead to much compression, which would translate into very marginal gains on the overall compression ratio. While it is possible to investigate more sophisticated compression schemes, in particular lossy compression of network weights (though this opens the problem of having to solve a trade-off between increasing weight compression and maintaining compression performance), this is beyond the scope of our paper. Accordingly, our compression rates can be understood as (somewhat) conservative estimates that give (in our case fairly tight) upper bounds on compression performance. The topic of compressing network weights to achieve competitive compression ratios would be of greater significance in a regime where models are significantly larger than ours (but the evaluation data stays roughly at the same size).

### B.4 SCALING ANALYSIS FOR MULTIMODAL TRAINING

Fig. A1 shows the results of simultaneously scaling dataset- and model size across training. In contrast to the similar Fig. 4 in the main paper, where models were trained on unimodal data, Fig. A1 shows models trained on multimodal data (i.e., the uniform mixture across all three modalities, with 55GB per modality). The multimodal training mixture acts as a regularizer, which can clearly be seen by the lack of overfitting of the largest models on images. Compare this against the unimodal training results in Fig. 4 where overfitting can be observed. In line with our other main results in Fig. 2 and Fig. 3, the overall compression ratios are slightly worse for the models trained on multimodal data compared to unimodal training.

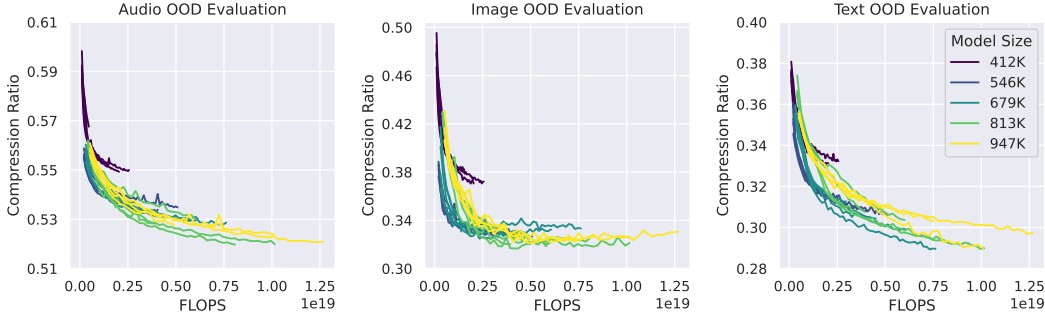

Figure A1: Similar to Fig. 4 in the main paper, but here the models are trained on a uniform mixture over all three modalities (55GB per modality). The plot shows compression performance evaluated on the unimodal datasets as training progresses for various model- and training set sizes (models are different colors, each line is a different training set size of either $20\%$, $40\%$, $60\%$, $80\%$, and $100\%$). We always train for 2 epochs, regardless of dataset size, i.e., smaller datasets require fewer FLOPS. In contrast to Fig. 4, where models are trained on unimodal data, we observe no overfitting, e.g., on images, even for the largest models tested. Note, however, that the compression ratios are slightly worse than for unimodal training, which is in line with our other expriments that show small losses when training on multimodal data.

