# OpenReview forum: "Compression via Pre-trained Transformers: A Study on Byte-Level Multimodal Data"
_ICLR.cc/2025/Conference — Submitted to ICLR 2025_

### Official Review · Reviewer_NUT8 · 2024-11-04

**Soundness:** 3
**Presentation:** 3
**Contribution:** 2
**Rating:** 5
**Confidence:** 3

**Summary:**

The paper outline a study of transformers / next-token-prediction for lossless compression of multiple modalities.
It focuses on relatively small models and show good results, across a variety of tasks and hyper-parameters.
The resulting model if able to compete and beat general and domain specific algorithms (FLAC for instance) on the different datasets and modalities.
One limitation with the model/trainset size is the lack of cross-domain compression abilities of these models, as remarked by the authors, and as somewhat expected.

**Strengths:**

I take this paper at face value. A lot of the conclusions of this paper are somewhat expected. Ability of next-token-prediction transformers to compete (and even beat) classic lossless compression algorithms, but also lack of abilities to move across domains, due to the limit in the parameters count / train-set size.

On the other hands this is also the strength of the paper - a clear claim and a credible set of results to substantiate such claims. Also results that can be seen as impactful.

**Weaknesses:**

I don't see any particular weakness, beyond the simplicity of this paper claims.
I am lent to ask the question about novelty, I'd like to say that perhaps there is no single paper coming to this conclusion, but there are bits and pieces in the literature pointing at exactly this conclusion. That is why I am not surprised by the results obtained in this work.

**Questions:**

- it maybe good to compare / scale the flops versus the classic algorithm we are comparing against ... it's possible that transformers are order of magnitude more expensive, almost surely, but would still be a good graph to have
- Figure-5, it's unclear: it seems to show compression ratio going up with model size (for audio) but down with context size (for video) ??
  I am unclear why model-size would at anytime make a consistent degradation in compression

---

> ### Author Response · Authors · 2024-11-20
>
> We thank the reviewer for their helpful feedback.
>
> **Can you compare the compressors in terms of the FLOPS required? Is it possible that transformers are an order of magnitude more expensive?**
>
> Computing the FLOPS for standard compressors (e.g., gzip or PNG) is much more involved than for neural networks and requires intricate knowledge of the algorithm, which can consist of more than 100K lines of code (also, note that standard compression algorithms have been highly optimized and run on CPU, whereas the neural compressors need a GPU). Instead, we compared the running times for all the compressors in Table A2. As the reviewer correctly suspected, neural compressors (i.e., both ours and those of Bellard (2021)) are orders of magnitude slower than standard compressors. Note that this does not undermine the relevance of our results, since we clearly stated that “the goal of our study is not to build a practical transformer-based universal compressor to compete with standard compressors in terms of computational footprint” in the limitations paragraph of Section 6.
>
> **What is surprising about the results of this work?**
>
> The most surprising finding is that small pre-trained transformers manage to compete with, and even outperform, standard domain-general and domain-specific compression algorithms *in the small(-ish) data regime*. This result does not trivially follow from previous work (Delétang et al., 2024), which shows that large models are strong compressors *without accounting for their parameter count* (i.e., they have low log-loss). Bellard's work provides perhaps the closest indication that our results may be achievable, but it operates in a very different paradigm (online training on the data to compress, whereas we pre-train and evaluate on out-of-distribution data) with a larger transformer. It could have equally been the case that small models (in our range of 20M parameters or less) do not manage to capture sufficient statistical structure to compete with algorithms designed to optimally exploit certain types of statistical structure.
>
> The other surprising aspect is that we see very little transfer to unseen modalities, even when training on multiple modalities, in contrast to what was observed with large-scale transformers (Delétang et al., 2024). One could imagine that small models learn very simple statistical patterns that are likely to transfer to other domains, whereas large models, trained only on text, learn complex and intricate domain-specific statistical patterns that do not generalize. However, the opposite seems to be true, and finding out whether this is mainly a question of model scale or dataset composition is an interesting future research direction. Our work aims to study transformers' domain-general inductive biases, and it seems that they are data- and scale-dependent (this, of course, is not an entirely novel finding, but our work adds an interesting and novel angle to this question).
>
>
> **Why does the compression ratio *increase* with increasing model size for audio in Figure 5 but not for images and text? Why would it ever increase with model size?**
>
> Figure 5 shows that for audio the compression ratio increases with increasing model size, but only if the context size is too small. We believe that this is because larger models overfit to the limited information available in short contexts, i.e., there are not enough complex dependencies that a large model could exploit. In contrast, text has a much higher information density than raw audio data, which is why Figure 5 shows that the compression ratio decreases with increasing model size even for the smallest context sizes we investigated. Note that, for large enough context sizes (i.e., 8192), the compression ratio on audio data decreases with increasing model size (for the sizes we investigated).

---

### Official Review · Reviewer_s6Zd · 2024-11-06

**Soundness:** 2
**Presentation:** 4
**Contribution:** 3
**Rating:** 6
**Confidence:** 4

**Summary:**

The paper proposes a pretrained Transformer architecture for doing compression on Byte level. They use simple vanilla transformer based architecture to carry this out. The findings are great, and have shown an expected results that transfer to unseen modalities is generally weak for cross modal compression on byte level data. The paper claims to have shown to achieve lowest compression ratio of 0.49 on OOD audio data vs other traditional audio compressors e.g. FLAC. The paper is well written, using a simple straightforward approach and its findings can perhaps lay the foundation to build on further complex architectures as well building byte level compressors that utilizes minimum domain specific knowledge.

**Strengths:**

strengths of paper on originality, quality, clarity, and significance

== Decent paper.
I wish the authors would have tweaked and added some novelty or some innovations for the compression than vanilla Transformer architecture. It would have made the paper really strong.

== I liked the idea of incorporating out-of-domain metrics for each of the modalities in addition to the out of modality benchmark

== The paper is well-written and explains all of the ideas proposed clearly. The experiments are well structured and the expectations are laid out at the very outset.

== The paper is reproducible.

== There is a good amount of insights from the paper that can be used for further studies or lay the foundation for future work and research directions.

== Simplicity. The paper is fairly simple and easy to read.

**Weaknesses:**

= Straightforward methodology. In brief, the paper compares vanilla transformers on byte level data for audio, text, images without any prior knowledge. What would have happened if we were to tweak the Transformer with something like Perceievers that also operate on Byte level.

== It would have been great if the authors would have evaluated atleast on one of the benchmarks the comparison with tokenization methods in text, audio and images. e.g. ENCODEC for audio and Byte-Pair Encoding for text

== These problems are quite different -- compression of 1024 or 8192 for audio at byte level is far simpler than compression of byte level data of text. The authors should have shed light on all of these different aspects, why it is and how it affects the compression ratios.

== The authors should have run experiment on scale of the number of parameters on experiment similar to Figure 3. What happens if I do one of these cross modal experiments on say a GPT-2 small or large level parameter architecture. Does it help ?  Does the gap narrow down. How does scaling affect the results.  It seems that the authors did have plenty of compute available for carrying out all of these experiments. Currently the scaling experiments are carried out on tiny Transformer models with maximum of 20M parameters.

== The methodology is straightforward. There could have been some novelty that could have been incorporated into the paper.

**Questions:**

Here are a few questions that would help me better evaluate the paper. The authors need not do any more experiments but rather simply address


== What would be the comparison with other advanced architectures such as perceivers. Would it help. If yes, why.  If not why?

== Why was there no comparison or experiment carried out for vanilla byte level architecture vs a tokenized in anyone of the three domains.

== Would the current results hold if we were to replace the input in all of the modalities by their respective token level representation. Which domains are likely to be improved the most.

== In figure 3 why are the compression ratios not getting better with size of the data. One should expect the plot to be similar to that of Figure 4.

---

> ### Author Response · Authors · 2024-11-20
>
> We thank the reviewer for their detailed and positive comments.
>
> **What would have happened if we were to tweak the transformer with something like Perceivers that also operate on byte level? Would it help? Why or why not?**
>
> As `Tvj5` correctly points out, raw (i.e., untokenized) data typically results in very long sequences (with lower information density). Thus, we would expect architectures that are optimized for that setting, such as the Perceiver (Jaegle et al., 2021) or Structured State Space Models (Gu et al., 2022), to improve the compression performance as they can exploit long-range dependencies more efficiently. However, note that, as stated in Section 6, our goal was “not to build a practical transformer-based universal compressor” but instead to “study *transformers’* learned inductive biases, e.g., whether they are domain-general, how they depend on the training data composition, and whether there is transfer between modalities” (see Section 1). We therefore did not investigate other model architectures, but we consider this a very promising avenue for future research, and we believe that alternative architectures that outperform vanilla transformers on our benchmark may have very interesting general inductive biases. We have updated the limitations section of our revised manuscript accordingly.
>
>
> **Would the current results hold if we were to replace the input in all of the modalities by their respective token-level representation. Which domains are likely to be improved the most?**
>
> We refer to our general response for a discussion on tokenization as pre-compression and note that there is no canonical token-level representation for a modality and that finding the optimal tokenizer for a single model/dataset combination is often the result of years of research, which is exactly why we keep our models domain-general by processing raw bytes.
>
> In the case of text data, Delétang et al. (2024) showed that the optimal tokenizer (raw bytes vs. byte pair encoding with vocabularies ranging from 1K to 20K) depends on the model size and that the compression ratios only improve by at most 3.6% (see Table 5 in their paper). Thus, we can safely assume that for text data the current results also hold if we were to replace the inputs with their respective token-level representations — with an error margin of a few percentage points. For other data modalities, the effects of tokenization for the purposes of lossless compression are unclear and an interesting direction for future research.
>
>
> **Why do the compression ratios in Figure 3 not improve with training dataset size (as in Figure 4)?**
>
> The compression ratios in Figure 3 *do* improve with increasing training dataset size (as in Figure 4), however, often not significantly. We hypothesize that this is because even the smallest training dataset is already sufficiently large to achieve competitive compression ratios and larger dataset sizes only marginally improve the compression because our models are too small to make good use of more data (larger models would not be competitive in terms of parameter-adjusted compression ratio).
>
> Note that all points of the same color correspond to the same amount of *total* training data (33GB, 66GB, 99GB, 132GB, and 165GB), and, for example, the left-most gray data point corresponds to 11GB of audio, 11GB of images, and 11GB of text while the left-most red data point corresponds to 33GB of text).
>
>
> **Could you shed light on how compressing 1028 or 8192 bytes for audio is far simpler than for text and how it affects the compression ratios?**
>
> Yes, Figure 5, where we ablate the model- and context size, sheds light on this phenomenon: For text, given a certain FLOPS budget, we always achieve the best performance with the largest model, i.e., the shortest context size (1024), which means that the model’s pattern-recognition capability (i.e., its size) is more important than being able to exploit long-range dependencies. For audio, we observe the opposite – given sufficient FLOPS, the best performance is achieved by the largest context (8192), i.e., the smallest model. We hypothesize that this is because the information density for raw audio is lower than for text (as `Tvj5` pointed out). Note that the best theoretically possible compression ratio depends on the redundancy of the raw data vs. the Shannon entropy of the data (the minimally achievable size). While the gap between these two determines the maximal compression ratio, it does not necessarily say how difficult it is to achieve this ratio.

---

### Official Review · Reviewer_c7rm · 2024-11-08

**Soundness:** 3
**Presentation:** 3
**Contribution:** 2
**Rating:** 6
**Confidence:** 4

**Summary:**

The paper conducts an empirical study to investigate the effectiveness of small pre-trained transformers (millions of parameters) as competitive multimodal compressors for text, image, and audio data. By training on a substantial dataset of 165GB per modality, the authors assess the compression performance on out-of-distribution (OOD) data for each modality. The results presented in the paper show that small transformers can outperform general-purpose and domain-specific compression algorithms, with compression ratios competitive even with state-of-the-art online adaptive transformers. Additionally, the paper explores the impact of multimodal training on compression efficacy, revealing that cross-modal transfer remains weak in these smaller models, contrasting with previously reported results for large-scale language models.

**Strengths:**

1. I find the paper to be thorough in its investigation of transformer-based compression across different modalities, with detailed experiments on model and context sizes, as well as dataset combinations.
2. The authors demonstrate that small, pre-trained transformers can achieve compression rates comparable to both domain-specific and general-purpose algorithms. Additionally, the paper shows that training on multiple modalities does not significantly impact unimodal performance while producing effective domain-general compressors.

**Weaknesses:**

1. The evaluation on a fixed 1GB test set limits the scalability of the compression methods. Did the authors explore test set sizes beyond 1GB or consider varying test data sizes? What is the rationale behind taking test data size as 1GB?
2. I noticed that the paper shows limited cross-modal transfer, struggling to compress modalities they weren’t trained on (e.g., training on text and audio did not generalize well to image data). Did the authors try any transfer learning approaches to address this?

**Questions:**

I have addressed all my questions in the weakness section.

---

> ### Author Response · Authors · 2024-11-20
>
> We thank the reviewer for their careful study of our paper and their insightful feedback.
>
> **Did you explore test set sizes beyond 1GB or consider varying test data sizes? What is the rationale behind taking test data size as 1GB?**
>
> We refer to our general response for a discussion of the test set size and discuss the additional ablation we conducted (Figure 6 in the revised manuscript) here:
>
> We agree that it is interesting to investigate varying the test data size, which is why we conducted an ablation for our best-performing models in Figure 6 of the revised manuscript. The figure clearly shows that for offline (i.e., pre-) trained neural compressors, for which the model parameters have to be factored into the compression ratio, the compression performance improves with increasing evaluation data (as long as the model generalizes well to the additional data). In contrast, the size of standard compressors is negligible compared to the amount of evaluation data, which means that their compression ratios are largely unaffected by the evaluation dataset size. Note that FLAC cannot losslessly compress more than 4.2GB of data.
>
>
> **Did you try transfer learning approaches to address the limited cross-modal transfer?**
>
> No, we did not investigate transfer learning as our goal was to “study transformers’ learned inductive biases, e.g., whether they are domain-general, how they depend on the training data composition, and whether there is transfer between modalities” (see Section 1), and, therefore, we did not deviate from the (tried-and-tested) standard supervised learning setup. Nevertheless, as you point out, it seems reasonable to suspect that transfer learning could potentially lead to considerable performance improvements. We have mentioned this as an interesting follow-up area in the limitations section of our revised manuscript.

---

> > ### Comment · Reviewer_c7rm · 2024-11-26
> >
> > Thank you for addressing my concerns. I now have a much clearer understanding of the experimental design used in the paper. I will maintain my original scores.

---

### Official Review · Reviewer_Tvj5 · 2024-11-09

**Soundness:** 3
**Presentation:** 4
**Contribution:** 2
**Rating:** 3
**Confidence:** 4

**Summary:**

This paper studies the connection between auto-regressive transformers and data compression. The authors presented an empirical study, investigating the usage of transformer-based models for data compression considering multi-modal data. The authors compared the transformer-based compressors with standard compression techniques while considering compressor sizes in comparison ratio calculations.
The authors show that small-scale transformer models can achieve better compression ratios than general multi-purpose compressors for in-domain modality data, however when considering out-of-domain modality data, standard compressors achieve superior performance.

**Strengths:**

1. This paper conducts a comprehensive empirical study on compression performance, especially for small transformer models considering multi-modal data. The paper is clearly written and easy to follow.
2. The presented results are interesting (the fact that small pre-trained transformers can achieve better compression ratios than standard general-purpose compressors) and would be of value to the community.
3. The authors provide scaling analysis on the performance of the models.

**Weaknesses:**

The main weakness of this study is its novelty / contribution. Considering the work done in [1,2], the contribution of this study is unclear. I understand that the authors additionally evaluated model performance under the multi-modal setup, however I'm afraid this is not enough for an ICLR publication.


[1] Bellard, Fabrice. NNCP v2: Lossless data compression with transformer. Technical report, Amarisoft, 2021.
[2] Delétang, Grégoire, et al. "Language modeling is compression." arXiv preprint arXiv:2309.10668 (2023).

**Questions:**

1. Following the weakness mentioned above, I would like to ask the authors to clarify their novelty/contribution to this research work.
2. How did you represent the multi-modal data? Representing the raw data can result in extremely large sequences. For example, music data is sampled at 48kHz, so only 10 seconds of music is a sequence length of 480k tokens. Similarly, how did you flatten the images?
3. Can the authors provide more details about model training and evaluation? How long did you train the model? On what sequence length? What was the sequence length during the evaluation? Etc.

I'm willing to change my score in case I missed something.

---

> ### Author Response · Authors · 2024-11-20
>
> We thank the reviewer for their constructive feedback.
>
> **How did you represent the multi-modal data? How did you flatten the images?**
>
> In Section 4 we stated that “we do not use tokenization and instead feed our models directly with byte streams” and that “Appendix A.1 describes the datasets in full detail”. In Appendix A.1 we then elaborated on the data representation for each modality:
> * *Audio*: “We downsample the dataset from 48 kHz to 16 kHz and encode the waveform as int16, i.e., with two bytes per sample.”
> * *Images*: “We decode the images into RGB arrays (three uint8 channels), flatten them (using `tf.reshape`), and concatenate them into a byte stream of flattened images.”.
> * *Text*: “UTF-8 encoded text”
>
> We are happy to include any other details the reviewers consider important.
>
>
> **Can you provide more details about training and evaluation? How long did you train the model? On what sequence length? What was the sequence during evaluation?**
>
> As stated in Section 4: “Unless otherwise noted, we use […] a context size of 4096 (bytes), and sliding windows without overlap or memory (full details in Appendix A.3). We train our models with the Adam optimizer (Kingma & Ba, 2015) for 2.5 million steps with a batch size of 32, which, for 165GB of data, roughly corresponds to 2 epochs.” We clarified that we always train and evaluate with the same sequence length in the revised manuscript.
>
> We are happy to include any other details the reviewer may have in mind.
>
>
> **Raw data can result in extremely large sequences. For example, music data is sampled at 48kHz, so only 10 seconds of music corresponds to 480K tokens.**
>
> Indeed, tokenization is a form of pre-compression (Delétang et al., 2024; Lester et al., 2024), mainly aimed at increasing the information density of the context window (at the cost of increased entropy), but the flip side is that raw data offers excellent potential for compression.
>
> In lossless compression, the ultimate limit is the Shannon entropy of the data source. Accordingly, if the raw data is much larger than this limit, it *must* contain a lot of redundancy. Whether a particular model can exploit this redundancy is a separate question (and, indeed, many domain-specific tokenizers target exactly this “trivial” redundancy), but large (w.r.t. redundancy) raw sequences are not a problem per se. In our setting, with a limited context window and alphabet (bytes as tokens), we can fit less information into the context, limiting our capability of exploiting long-range dependencies in the data (which is why large language models benefit from tokenization and larger alphabets). At the same time, highly pre-compressed data makes learning to predict/compress harder, and compression gains are generally smaller (Lester et al., 2024). In practice, this leads to an interesting and non-trivial design space where we can explore model size, tokenization, and data modality. We did not engage with this problem to avoid complicating the main findings and takeaways, and we refer to Delétang et al. (2024) for an analysis of the interplay between tokenization and compression.
>
> Also, note that we (sub-) sample our audio data at 16kHz, not 48kHz (see Figure 1 and Appendix A.1).

---

> > ### Comment · Reviewer_Tvj5 · 2024-11-23
> > **Official Comment by Reviewer Tvj5**
> >
> > I would like to thank the authors for providing additional details and clarifications for my questions. I better understand the authors research question. However, I would like to keep my score unchanged.

---

> > > ### Author Response · Authors · 2024-11-25
> > >
> > > The reviewer stated: `I'm willing to change my score in case I missed something`. Given that the review missed substantial aspects of our work (see below), we kindly ask them to revisit their score.
> > >
> > >
> > > **Novelty**
> > >
> > > The main weakness pointed out by the review is `novelty/contribution [...] considering the work done in [1, 2]`. We have stated the novelty and contribution of our work and clarified how it relates to [1] and [2] in the general response. ***We thus refer to the general response and provide further details on the review’s misunderstandings below.***
> > >
> > > The review states that our only contribution over [1, 2] is to `[evaluate] model performance under the multimodal setup`. ***That is incorrect***:
> > >
> > > ***In contrast to [1]***, we use *offline* adaptive arithmetic coding, which induces entirely *different constraints on the optimal model size* (since the model parameters do not have to be factored into the compression ratio). [1] relies on "test-time" gradient updates on in-distribution data, whereas we pre-train models and then leverage in-context learning on out-of-distribution data. ***It is, therefore, impossible to conclude from the results in [1] whether our results are possible.*** The training data and protocol are incomparable, making it all the more interesting that our results are generally very close — another surprising finding that cannot be trivially explained.
> > >
> > > ***In contrast to [2]***, we do not evaluate off-the-shelf, large-scale, text-based foundation models but pre-train small-scale transformers on audio, image, and text data. As a result, the compressors proposed by [2] are not competitive w.r.t. standard compressors (which they acknowledge) — unlike our models (we beat standard compressors across the board). [2] do conduct a pilot experiment by pre-training a small transformer on text data, but, in contrast to our work, they do not perform (i) a comprehensive study of multi-modal training, (ii) out-of-distribution evaluation, (iii) ablations over context-, dataset-, and model-size, (iv) a comparison to [1], (v) an investigation of the sliding window overlap, and (vi) a test dataset size ablation. As our work shows, *all* of these ingredients are necessary to obtain the strong performance we reported, and, as a result, ***[2] do not manage to pre-train a transformer-based compressor that beats standard compressors across multiple modalities — unlike our work.*** The main contribution of [2] is conceptual, with a relatively simple experimental evaluation to illustrate the conceptual arguments. Our work, on the other hand, performs a rigorous and comprehensive empirical study.
> > >
> > > We are happy to add these clarifications to the paper if the reviewer finds them helpful.
> > >
> > >
> > > **Missing Details**
> > >
> > > The *only* other weakness identified by the review is the lack of details regarding the data representation and model training/evaluation. As pointed out in our response above, ***all of these details were already included in the paper, which the review missed***.
> > >
> > >
> > > **References**
> > >
> > > [1] Bellard. NNCP v2: Lossless Data Compression with Transformer. 2021.
> > >
> > > [2] Delétang et al. Language Modeling Is Compression. 2024.

---

> > > > ### Comment · Reviewer_Tvj5 · 2024-11-25
> > > > **Official Comment by Reviewer Tvj5**
> > > >
> > > > I would like to thank the authors for the additional clarifications. As I stated before, I better understand the author's research question, however, I believe it does not pass the bar for an ICLR publication. Hence, I would like to keep my score unchanged.

---

### Author Response · Authors · 2024-11-20
**General Response**

We thank the reviewers for their positive feedback and are pleased that they consider our
* results impactful and of value to the community (`Tvj5`, `c7rm`, `NUT8`)
* experiments comprehensive, well-structured, and reproducible (`Tvj5`, `c7rm`, `s6Zd`)
* paper well-written (`Tvj5`, `s6Zd`) with a clear claim and a credible set of results to substantiate it (`NUT8`)
* work as laying the foundation for future research (`s6Zd`)

Here, we summarize our response to the common questions. We reply to the individual questions below every review.

**Additional Experiment** We studied the effect of compressing smaller/larger datasets in Figure 6 as requested by `c7rm`.

**What is the contribution/novelty over [1, 2]?** (`Tvj5`, `s6Zd`, `NUT8`)

| | # of Params | Training | Multimodal | Adjusted Compression Ratio |
|---|:---:|:---:|:---:|:---:|
| [1] | 187M | online | no | yes |
| [2] | 70B | offline | yes | no |
| Ours | 20M | offline | yes | yes |

As stated in Section 1 and shown in the table above, we study the *open* question of whether small transformers pre-trained on multimodal data can achieve competitive compression ratios across different modalities and whether there is transfer to unseen modalities (as observed in the large-scale model case). Consequently, the fact that they can do so is *surprising and novel* and cannot be predicted from the strong compression performance of LLMs (e.g., results by [2]).

Unlike [1], we use offline (i.e., pre-) training and consider multimodal data. Online training imposes different constraints on the optimal model size, since the compression ratio does not include the model parameters, which is why [1] studied much larger models. We also conduct comprehensive scientific ablations over dataset-, model-, and context-size.

Unlike [2], we (only) consider the *adjusted* (i.e., the actual) compression ratio, which includes the model parameters, and therefore, we cannot use massive foundation models. Moreover, while [2] studied the cross-domain transfer of foundation models trained on text, they did not study multimodal training.

**Why do you not investigate larger models?** (`c7rm`, `s6Zd`)

Since we study the offline (i.e., pre-trained model) regime, our compression ratios include the model size. When ignoring parameter size, large-scale foundation models achieve the best compression (i.e., lowest log-loss), which is, perhaps, not very surprising and what [2] showed. In contrast, when accounting for parameter size, the evaluation data size plays a pivotal role — if the model size is the same as or larger than the data size, no amount of compression can lead to a compression ratio below 1. Thus, to compete with standard compressors, models have to be an order of magnitude smaller than the data. We can achieve this by either increasing the data size (not very interesting since standard compressors are optimized for small files and would be outperformed by LLMs that capture more higher-order and long-range statistical structure) or by decreasing the model size, which is more challenging and exactly what we focus on.

**Why do you compress 1GB of evaluation data?** (`c7rm`, `s6Zd`)

1GB is a good compromise where large-scale models are infeasible (with more data standard compressors are easier to beat) but the permissible models can still be pre-trained into strong predictors (with less data good neural predictors are unlikely). It is also the standard size in text compression benchmarks (http://prize.hutter1.net, https://www.mattmahoney.net/dc/text.html). We ***added an ablation over the dataset size to the revised manuscript*** to empirically show this trend.

**Why do you not use tokenizers (e.g., EnCodec for audio or byte pair encoding for text)?** (`Tvj5`, `s6Zd`)

Tokenization is a *domain-specific* pre-compression step aimed at eliminating “simple” redundancies. It is also an active research area, and there is no obvious choice for a multimodal, let alone domain-general, tokenizer. While we consider tokenization an important problem in practice, for our study, it makes evaluations more complicated with little further insight. Our goal is to study transformers’ compression performance and not the (pre-) compression performance of various tokenizers.

For models with fixed context size, e.g., transformers, the tokenizer's pre-compression increases the information density in the context, which helps to capture long-range correlations. However, it also makes the learning problem harder since the data has higher entropy. This leads to an intricate design space, and we chose not to complicate our results with these considerations (but they are relevant in practice).

Note that EnCodec [3] is a lossy compressor and, thus, cannot be used as a tokenizer in our work.

**References**

[1] Bellard. NNCP v2: Lossless Data Compression with Transformer. 2021.

[2] Delétang et al. Language Modeling Is Compression. 2024.

[3] Défossez et al. High Fidelity Neural Audio Compression. 2023.

---

### Meta-Review · Area_Chair_1oEP · 2024-12-22

**Metareview:**

> This paper studies the connection between auto-regressive transformers and data compression. The authors presented an empirical study, investigating the usage of transformer-based models for data compression considering multi-modal data. The authors compared the transformer-based compressors with standard compression techniques while considering compressor sizes in comparison ratio calculations. The authors show that small-scale transformer models can achieve better compression ratios than general multi-purpose compressors for in-domain modality data, however when considering out-of-domain modality data, standard compressors achieve superior performance.

Reviewer Tvj5's concerns were partially addressed by the authors in the rebuttal, but not enough to convince them to change their score. Nevertheless, even discounting heavily review Tvj5, the paper scored 6-6-5.

The paper doesn't cite or compare to Lossless Data Compression with Transformer (Izacard et al.) https://openreview.net/pdf?id=Hygi7xStvS nor to MEGABYTE: Predicting Million-byte Sequences with Multiscale Transformers (Yu et al.) https://arxiv.org/abs/2305.07185 . Previous work studying transformers, or simply language models for lossless compression, have discussed all of the claims.

Overall, the paper doesn't seem to provide a sufficiently new insight, that is not known by the community of ICLR. Simplicity overall is good, and novelty for novelty's sake is not the point, but none of the reviewers noted that the framing of the paper taught them something.

**Additional Comments On Reviewer Discussion:**

Reviewers Tvj5 and c7rm took part in the rebuttal discussion.

---

### Decision · Program_Chairs · 2025-01-22

Reject